# Oil Extraction Systems Influence the Techno-Functional and Nutritional Properties of Pistachio Processing By-Products

**DOI:** 10.3390/foods14152722

**Published:** 2025-08-04

**Authors:** Rito J. Mendoza-Pérez, Elena Álvarez-Olmedo, Ainhoa Vicente, Felicidad Ronda, Pedro A. Caballero

**Affiliations:** 1Department of Agriculture and Forestry Engineering, Food Technology, College of Agricultural and Forestry Engineering, University of Valladolid, 34004 Palencia, Spain; ritojose.mendoza@uva.es (R.J.M.-P.); elena.alvarez.olmedo@uva.es (E.Á.-O.); ainhoa.vicente@uva.es (A.V.); 2Research Institute on Bioeconomy-BioEcoUVa, PROCEREALtech Group, University of Valladolid, 47011 Valladolid, Spain

**Keywords:** partially defatted pistachio flour, pistachio by-product, pistachio oil, functional properties, hydraulic press, single-screw press

## Abstract

Low-commercial-value natural pistachios (broken, closed, or immature) can be revalorised through oil extraction, obtaining a high-quality oil and partially defatted flour as by-product. This study evaluated the techno-functional and nutritional properties of the flours obtained by hydraulic press (HP) and single-screw press (SSP) systems, combined with pretreatment at 25 °C and 60 °C. The extraction method significantly influenced flour’s characteristics, underscoring the need to tailor processing conditions to the specific technological requirements of each food application. HP-derived flours presented lighter colour, greater tocopherol content, and higher water absorption capacity (up to 2.75 g/g), suggesting preservation of hydrophilic proteins. SSP-derived flours showed higher concentration of protein (44 g/100 g), fibre (12 g/100 g), and minerals, and improved emulsifying properties, enhancing their suitability for emulsified products. Pretreatment at 25 °C enhanced functional properties such as swelling power (~7.0 g/g) and water absorption index (~5.7 g/g). The SSP system achieved the highest oil extraction yield, with no significant effect of pretreatment temperature. The oils extracted showed high levels of unsaturated fatty acids, particularly oleic acid (~48% of *ω-9*), highlighting their nutritional and industrial value. The findings support the valorisation of pistachio oil extraction by-products as functional food ingredients, offering a promising strategy for reducing food waste and promoting circular economy approaches in the agri-food sector.

## 1. Introduction

The pistachio tree (*Pistacia vera* L.), native to the Middle East, is now widely cultivated across the Mediterranean basin and other regions with a similar climate due to its remarkable capacity to adapt to environmental conditions [1,2]. Global pistachio production has grown steadily, reaching 1,303,461 tonnes in 2023, with Iran, the United States, Turkey, China, and Syria leading global production, while Italy, Greece, and Spain are the principal producers within Europe [3].

Pistachio nuts are valued not only for their sensory qualities but also for their nutritional composition, which contributes to their increasing consumption. They contain approximately 45–62% lipids, characterised by a high proportion of unsaturated fatty acids, mostly oleic (51–81%), linolenic (8–31%), and palmitic acids (7–15%) [1,4], alongside about 8–12% dietary fibre, 27–28% carbohydrates, and 18–22% protein [4,5]. Furthermore, pistachios are rich in vitamins, minerals (including notable amounts of calcium, magnesium, phosphorus, and potassium), and bioactive compounds such as tocopherols, phytosterols, and phenolic compounds. The beneficial effects associated with pistachio consumption are closely linked to their nutritional composition, which has been shown to contribute to enhanced cardiovascular health, reduced cholesterol levels, and a decreased risk of certain cancers, in addition to providing antioxidant properties [6,7,8].

The increasing demand for pistachios has given rise to a growing interest in the valorisation of by-products generated during their processing. Pistachios are primarily consumed directly as an aperitif, either natural or roasted and salted [2], although they are also used in confectionery and in the elaboration of products such as spreads or vegetable drinks [8]. However, there is a fraction of pistachios that does not meet the quality requirements for direct consumption, including damaged, broken, stained, or immature kernels [8]. For these kernels, oil extraction has emerged as an interesting process to valorise them, due to the high lipid content and favourable fatty acid profile of the nuts [9,10,11]. Pressure systems, including hydraulic and screw presses, are commonly employed in the extraction of nut oils due to their cost-effectiveness and ability to produce high-quality oils [9,11]. Importantly, these extraction processes generate a partially defatted flour as a by-product. While traditionally used in animal feed [12], this flour represents a potentially valuable ingredient for human food applications. Its interesting nutritional profile makes it suitable for the development of new functional foods [9], bakery products [13], and as an alternative to wheat flour in gluten-free products [14].

Despite this potential, the scientific literature on the characterisation and utilisation of pistachio oil extraction products and by-products remains limited. Most studies to date have focused primarily on the oil, examining aspects such as yield from different pressure systems, and sensory and physicochemical properties, including acidity index, peroxide value, oxidative stability, fatty acid composition, and polyphenol concentration [6,10,11]. Studies that have investigated the byproduct, the partially defatted flour, have largely focused on its nutritional characteristics, including proximal composition, mineral content, amino acid profile, bioactive compounds, and antioxidant properties [15,16,17]. However, there is a notable paucity of information on the techno-functional properties of these flours.

The functional properties of defatted flours have been studied in other raw materials such as oilseeds [18,19] and other nuts [20]. Fernández-López et al. (2018) [21] demonstrated that defatted chia flour exhibited high values of emulsion activity and stability, suggesting that they may play an important role as good emulsifiers and in stabilising oil-in-water emulsions in the food industry. Mendoza-Pérez et al. (2024) [19] concluded that the variety of hemp used to obtain defatted flour notably influenced its water absorption capacity, emulsion activity, and stability, which could be adapted for use in various food applications. As established by Burbano et al. (2024) [18], partially defatted cashew nut flour has been shown to exhibit high emulsifying activity and high water and oil absorption capacity. Other studies have examined the functionality of extracts from various nut components, including protein concentrates and isolates from different raw materials, such as cashew nuts [20] and conophor nuts [22,23]. Their results demonstrated the viability of using them as a high-quality protein ingredient in the production of ice creams, bakery products, and drinks due to its favourable foaming and emulsifying properties [20,21,22,23].

The aim of this study was to provide a comprehensive analysis of the techno-functional properties of defatted pistachio flours obtained as by-products of oil extraction. The influence of different oil extraction systems commonly used in the processing of nuts (hydraulic press and single-screw press) and pretreatment conditions of pistachios (initial sample temperatures of 25 °C and 60 °C) was investigated. Complementarily, the nutritional properties of both the partially defatted pistachio flours and the oils were characterised. The findings offer new insights into the technological potential of defatted pistachio flour as a functional ingredient for food applications, depending on the oil extraction conditions applied.

## 2. Methods

### 2.1. Natural Pistachios

The pistachios used in this study were natural deshelled kernels obtained from smaller-sized nuts or those of inferior quality (broken, closed, or immature). A representative batch of these kernels were provided by the company Pistacyl (Valladolid, Spain), who carried out a post-harvest drying to stabilise moisture content below 5%, mechanical shelling, and storage under a modified atmosphere (100% N_2_) prior to delivery. These pistachios were stored at 10 °C for two months prior to processing and analysis. The proximal composition of natural pistachios, expressed as % (*w*/*w*), was as follows: 4.1 ± 0.2% moisture, 46 ± 1% fat, 9 ± 2% fibre, 26 ± 2% protein, and 3.0 ± 0.4% ash.

The proximal composition of pistachio kernels and their derivatives was determined according to the following methods: moisture was analysed following the AACC Method 44-19.01 [24]; protein content was determined by the Kjeldahl method (AACC Method 46-10.01; conversion factor 6.25) [24]; lipid content was measured by solvent extraction using the Soxhlet method (AACC Method 30-25.01) [24]; ash content was determined with the AACC Method 08-01.01 [24]; and fibre content was analysed following AOAC Method 991.43 [25]. Total carbohydrate content was calculated by difference, subtracting the moisture, fat, protein, fibre, and ash content from 100%. Analyses were performed in duplicate.

### 2.2. Oil Extraction Process

Figure 1 summarises the overall oil extraction process employed in this study, including the thermal pretreatments and pressing methods applied to the pistachio samples.

#### 2.2.1. Thermal Pretreatment of Pistachios

Natural pistachios were sliced using a disc cutter (Halde RG-100, Oświęcim, Poland) with a blade spacing of 1.5 mm. The sliced samples were subjected to a thermal pretreatment at either 25 °C and 60 °C for 30 min in an incubator chamber (Memmert UN750, Schwabach, Germany).

#### 2.2.2. Single-Screw Press (SSP)

Pretreated pistachios were subjected to oil extraction using a single-screw press (SSP) (CZR 309, Goldenwall, Hangzhou, China) following the procedure of Mendoza-Pérez et al. (2024) [19]. In each trial, 200 g of pretreated pistachios were pressed. A digital thermometer (Testo 735-2, Testo Instruments S.A., Barcelona, Spain) was used to monitor the temperature during each extraction process, which did not exceed 80 °C. Extractions were performed in triplicate.

#### 2.2.3. Hydraulic Press (HP)

Alternatively, oil was extracted using a hydraulic press (HP) (Ibertest ELIB-100, Madrid, Spain). For this purpose, 15 g of pretreated pistachios were introduced into a 35 mm diameter extractor, subjected to progressive pressure up to 98 MPa, and held for 4 min, with a total extraction time of 8 min. The process was performed in triplicate.

#### 2.2.4. Oil and By-Product Processing

The pistachio oils obtained were centrifuged (7000× *g*, 30 min, 10 °C) to remove the remaining solids and stored in dark glass bottles at 4 °C until analysis. Oils were identified as PO-HP25 and PO-SSP25 (from pistachios pretreated at 25 °C using HP and SSP, respectively) and PO-HP60 and PO-SSP60 (from pistachios pretreated at 60 °C using HP and SSP, respectively).

The press cakes obtained as a by-product from the extraction process were ground using an electric coffee grinder (HC-400, Cgoldenwall, Hangzhou, China) for 30 s (three 5 s intervals interspersed with 5 s rest periods). The ground product was passed through a 500 μm sieve to obtain the partially defatted flours, which were stored at 4 °C until analysis. Flours were identified according to both the pistachio pretreatment and the extraction method used: PF-HP25 and PF-SSP25 (from pistachios pretreated at 25 °C using HP and SSP, respectively) and PF-HP60 and PF-SSP60 (from pistachios pretreated at 60 °C using HP and SSP, respectively).

### 2.3. Determination of Extraction Yields

The oil extraction yield was determined by calculating the mass of extracted oil or obtained cake relative to the initial mass of pistachio kernels (g/100 g).

### 2.4. Characterisation of Oil

The proximal composition was determined as described in Section 2.1.

Free acidity, expressed as % oleic acid, was determined by titration of an oil solution dissolved in ethanol/ether (1:1) with a 0.1 N ethanolic potassium hydroxide solution, using phenolphthalein as the indicator [26].

The fatty acid (FA) composition was determined according to AOAC Official Method 41.1.30 [25], with slight modifications. The FA methyl esters were analysed using a gas chromatography system (Agilent-6890N, Agilent Technologies, Santa Clara, CA, USA) equipped with an autosampler (7683B, Agilent Technologies, USA) and a flame ionisation detector (FID). Helium was used as carrier gas at a flow rate of 1.8 mL/min and a fused silica capillary column (Omegawax TM-320, 30 m × 0.32 mm, Agilent Technologies, Santa Clara, CA, USA). FA methyl esters were identified by comparing their retention times with those of chromatographic standards (Sigma-Aldrich, USA). Quantification was performed by relating the area of the peaks to the area of an internal standard (methyl tricosanoate), as indicated in AOAC Method 41.1.30. Calibration curves were prepared for pairs consisting of the internal standard and several representative chromatographic standards to determine the corresponding response factors.

The colour was evaluated by determining the CIEXYZ and CIELAB parameters, following the CIE 15:20004 method [27], with slight modifications. The XYZ coordinates of the samples were obtained from 30 transmittance measurements of samples diluted with hexane (1:4 *v*/*v*) using a spectrophotometer (Lambda 25, PerkinElmer, CA, USA) at selected wavelengths between 422.2 nm and 645.9 nm, according to the method. The lightness, *L**, ranging from 0: black to 100: white, and the chromatic coordinates *a**, ranging from green (−) to red (+), and *b**, ranging from blue (−) to yellow (+), were established from the XYZ values using illuminant C and 2° observer. The hue (h) and chroma (*C**) were calculated as described by Abebe et al. (2015) [28]. All measurements were performed in duplicate.

### 2.5. Nutritional Composition and Colour of Flours

The proximal composition of the partially defatted pistachio flours was determined in accordance with the procedures described in Section 2.1.

The mineral content was determined using the method of Ronda et al. (2015) [29], with slight modifications. Briefly, 0.5 g of flour was digested with 10 mL of 85% high-purity HNO_3_ using microwave technology (ETHOS SEL, Milestone, Italy). The digested sample was then diluted to 50 mL with distilled water and filtered through 0.45 μm with syringe filters. Calcium (Ca), potassium (K), magnesium (Mg), phosphorus (P), iron (Fe), and zinc (Zn) were quantified using an inductively coupled plasma optical emission spectrometer (ICP-OES) (VARIAN 725-ES, Agilent Technologies, Santa Clara, CA, USA). The assays were performed in duplicate.

The tocopherol content (mg/kg) of the residual oil contained in the partially defatted flours was determined according to the IUPAC 2.432 official method as reported by Rebolleda et al., (2012) [30] using HPLC (Agilent 1100, Agilent Technologies, Santa Clara, CA, USA) with a silica gel column and n-hexane/2-propanol mobile (1 mL/min flow rate). A fluorescence detector was used with excitation and emission wavelengths set at 290 and 330 nm, respectively. α-, γ-, and δ-tocopherols were identified and quantified using external calibration curves of the corresponding standards. Determinations were conducted at least in duplicate.

The colour was determined using a colorimeter (PCE-CSM5, PCE Instruments, Meschede, Germany) and CQCS3 software. The CIEL*a*b* coordinates were obtained using a D65 standard illuminant and a 10° standard observer. The hue (h) and chroma (*C**) were calculated from the CIEL*a*b* coordinates [28]. All measurements were performed five times.

### 2.6. Techno-Functional Properties of Flours

The techno-functional properties were determined in fully defatted flours. Defatting was performed in partially defatted flours using solvent extraction with Soxhlet according to the AACC Method 30-25.01 [24]. Complete defatting was necessary to eliminate interference from residual fat, which could affect the accuracy of functional property measurements in flours obtained by pressing.

The hydration properties of the flours, including water absorption capacity (WAC), water absorption index (WAI), water solubility index (WSI), and swelling power (SP), were evaluated following the method described in Calix-Rivera et al. (2023) [31]. Foaming properties, including foam capacity (FC) and foam stability (FS), were determined according to Abebe et al. (2015) [28]. Emulsifying activity (EA) and emulsion stability (ES) were assessed following Vicente et al. (2023) [32], with slight modifications. Briefly, 7 g of flour was mixed with 100 mL of water and 100 mL of corn oil (Koipe Asua, Córdoba, Spain). The mixture was homogenised for 60 s at 1000 rpm and distributed into 50 mL centrifuge tubes. The initial emulsion volume (V_1_) was recorded. The tubes were subsequently centrifuged at 1300× *g* for 5 min, and the volume of the remaining emulsified layer (V_2_) was measured. EA was calculated as V_2_/V_1_ and expressed as a percentage. To determine ES, the emulsions were heated to 80 °C for 30 min, cooled to room temperature, and centrifuged at 1300× *g* for 5 min. ES was expressed as the percentage of emulsion retained after heating. All measurements were performed in triplicate, and the results were expressed on a dry matter (dm) basis.

### 2.7. Statistical Analysis

The data were statistically analysed using Statgraphics Centurion 19 (Bitstream, Cambridge, MN, USA). Analysis of variance (ANOVA) and the Least Significant Difference (LSD) test at *p*-value < 0.05 was performed. Distinct letters were assigned within tables to denote statistically significant differences among means. For each parameter evaluated, the mean values (calculated from the number of replicates specified in earlier sections) are presented alongside the pooled standard error (SE) derived from the ANOVA. In addition, a multivariate analysis of variance (MANOVA) was applied to assess the combined effect of the experimental variables on the measured properties.

## 3. Results and Discussion

### 3.1. Extraction Yields

Table 1 shows the yields of oil and cake obtained under different extraction systems, single-screw press (SSP) and hydraulic press (HP), and pretreatment temperatures, 25 °C and 60 °C. The extraction system had a statistically significant impact on both oil and cake yield (*p* < 0.001), with the SSP achieved a markedly higher oil yield (approximately 40%) compared to the HP (approximately 25%). In contrast, the pretreatment temperature and its interaction with the extraction system did not significantly affect yields (*p* > 0.05). This was likely due to the mild pretreatment conditions used (30 min at 25 or 60 °C), whereas more intense pretreatment conditions have been shown to be effective in increasing extraction yields in other matrices [33].

Consistent with our findings, previous studies have reported significantly higher oil yields when using SSP systems compared to HP systems. Roncero et al. (2021) [9] observed average oil yields of 49.18% for SSP and 37.94% for HP in almond kernel extraction. Similarly, Sena-Moreno et al. (2015) [6] reported yields of approximately 40% for SSP and 32% for HP when processing pistachio kernels. In our study, the oil yield obtained using SSP was comparable to that reported by Sena-Moreno et al. (2015) [6], while the yield for HP was lower (25% vs. 32%). The higher efficiency of screw presses has been primarily attributed to two key factors: (1) the enhanced interaction between the mechanical components of the system and the material being pressed, where the rotational movement of the screw generates frictional shear forces that facilitate the rupture of parenchyma structures and oil-containing liposomes [9,34]; and (2) the higher extraction temperatures reached during screw pressing, will reduce oil viscosity and promote coalescence of oil droplets, thereby enhancing their diffusion towards the external surface of the cake and ultimately facilitating greater oil release and higher extraction yields [9,35].

### 3.2. Characteristics of Extracted Pistachio Oils

The colour parameters of the oils resulting from pistachio processing are shown in Table 1. The lightness (*L**) was significantly influenced by the extraction system (*p* < 0.01) and its interaction with the temperature (*p* < 0.05), but not by the temperature alone (*p* > 0.05). Notably, the oil extracted using SSP showed a darker colour, with a 21% reduction in *L** compared to the oil obtained with HP. This difference is likely attributable to the heating experienced by the sample during the screw press extraction process. The darker colour of the PO-SSP60 sample compared to the PO-SSP25 sample may also reflect the higher starting temperature, which, combined with the heat generated by the SSP, subjected the oil to greater thermal exposure. In contrast, no such effect was observed with HP, as this system does not significantly increase the product temperature during extraction [14]. The colour of the oil was located in the second quadrant of the chromatic diagram, with negative *a** values and positive *b** values, indicating a yellowish-greenish colour. The chromatic coordinates *a** and *b** were differently affected by the studied factors. The *a** coordinate was notably impacted by the pretreatment temperature, with samples treated at 60 °C displaying a significantly higher value (*p* < 0.01). The chromatic coordinate *b** was higher for the oils extracted using the HP than for those obtained with the SSP and was only affected by the pretreatment temperature in the SSP system, showing a reduction at the higher temperature. These modifications in *a** and *b** led to different variations in hue (h) and chroma (*C**). The highest h value, indicating a more greenish tone, was observed for the PO-HP25 sample and the lowest for the PO-HP60, highlighting the great effect of temperature pretreatment for HP extraction. The SSP system showed intermediate values, without variations due to pretreatment temperature. The *C** was higher for samples extracted using the HP system, without a significant effect of temperature (*p* > 0.05), indicating a more vivid colour. However, the pretreatment temperature did affect the *C** in the SSP system, with a 11% reduction for PO-SSP60 compared to PO-SSP25. Sena-Moreno et al. (2015) [6] reported greener oil colours when using SSP compared to HP for pistachio oil extraction, particularly at higher temperatures. This shift towards a greener colour was associated with greater consumer acceptability in sensory evaluations. Similar findings were reported by Rabadán et al. (2018) [1], who showed that pistachios oils obtained from raw or minimally roasted pistachios (50–75 °C) exhibit a yellow hue, whereas those extracted from pistachios roasted at higher temperatures (100–125 °C) displayed a bright green coloration. The observed colour shift, characterised by a reduction in *L** and *b** values and an increase in *a** value, was attributed to Maillard reactions induced by the roasting process and the degradation of chlorophylls [1,36]. In the present study, the preheating required for SSP operation may have similarly altered the colour of the pistachio to a darker one. This change is likely due to the accumulation of products from Maillard reactions, caramelization compounds, or phospholipid degradation. Furthermore, the elevated temperature and continuous friction generated during SSP may have accelerated these reactions, thereby intensifying the observed colour changes [34].

The free acidity values of the studied oils are presented in Table 1. All samples exhibited low free acidity levels of 0.6%, expressed as oleic acid, which were not significantly influenced by any of the studied factors, namely the extraction system employed and pretreatment temperature (*p* > 0.05). Similar values were reported by Sena-Moreno et al. (2016) [10] when comparing SSP and HS extraction of pistachio oil, supporting the observation that low acidity levels are indicative of greater freshness and overall oil quality [37]. For reference, virgin olive oil must have a free acidity below 0.8% to qualify as extra virgin [9], placing the pistachio oils obtained in this study well within the range considered optimal for high-quality edible oils.

Table 2 shows the fatty acid (FA) profile of the oil samples obtained using two extraction systems under the same pretreatment conditions (25 °C). The pretreatment temperature was not included as a study factor in the FA analysis, as previous studies on pistachio and grape seed oils have shown that processing temperature does not significantly affect fatty acid composition [6,10,37]. Therefore, this study focused solely on the influence of the extraction system on the FA profile.

Pistachio oils extracted using both pressing systems exhibited a fatty acid profile characterised by a low concentration of saturated fatty acids (SFAs) (15.1–15.6%) in comparison to monounsaturated fatty acids (MUFAs) (51–54%) and polyunsaturated fatty acids (PUFAs) (31–32%). Within the PUFAs, *ω-6* fatty acids were predominant (30–31%) compared to *ω-3* fatty acids (0.76–0.86%). *ω-9* fatty acids accounted for 48–49% of the total, representing 90.7–94.1% of MUFAs content. The major fatty acids identified were oleic acid (47–49%), followed by linoleic acid (30–31%) and palmitic acid (14%). No significant differences (*p* > 0.05) in fatty acid composition were observed between oils extracted with the different systems, confirming that the extraction method did not influence the FA profile. These findings align with previous studies on pistachio and other nut oils [6,38], which also reported negligible effects of the extraction system on fatty acid composition. The only significant difference observed in this study was the higher content of trans fatty acids (TFAs) in oils extracted using the SSP compared to HP (*p* < 0.001). This increase may be attributed to the higher temperatures reached during screw pressing, combined with the presence of considerable amounts of linoleic acid, which is highly susceptible to isomerisation, leading to the formation of trans-isomers [39]. Nevertheless, the measured TFAs content (2 mg/g) remained well below the regulatory limit established in some countries for edible oils, such as Denmark, which sets a maximum of 20 mg/g [39].

### 3.3. Nutritional Composition and Colour of Flours

#### 3.3.1. Proximal and Mineral Compositions

The proximal composition of the partially defatted pistachio flour samples obtained as a by-product of oil extraction is presented in Table 3. The findings indicate that the extraction system significantly influenced (*p* < 0.05) all parameters except carbohydrates, while the pretreatment temperature had no significant influence in any parameter (*p* > 0.05) except moisture (*p* < 0.001). However, the interaction of these factors had a significant effect (*p* < 0.05) on the moisture and fat content of the resulting flours.

The use of the SSP extraction system resulted in an average reduction of approximately 30% in moisture content compared to HP. This effect was further accentuated by the application of higher temperatures in the pretreatment step. The observed moisture reduction is likely due to combined drying effects: moisture loss during the thermal pretreatment and additional drying for SSP due to the heating of the screw press.

As a result of the prior oil extraction, the flours exhibited reduced lipid content compared to natural pistachio flours [40]. The extraction system used significantly influenced the fat content of the flours (*p* < 0.001), whereas the pretreatment temperature did not show a significant effect (*p* > 0.05). The extraction of oil from pistachios using the SSP system at pretreatment temperatures of 25 °C and 60 °C yielded flours with reduced fat content, 16.8 and 18.7 g/100 g, respectively. In contrast, flours derived from the HP system contained higher fat levels, approximately 28 g/100 g, with no significant differences between the two pretreatment temperatures. These findings were consistent with the previously reported lower oil extraction yields observed with HP (Section 3.1), which led to a higher residual oil content in the flour. This residual oil may cause a dilution effect on the concentration of other components, influencing the overall composition of the samples.

The extraction of a significant portion of oil from pistachios results in a by-product with an elevated protein content. Flours resulting from SSP system showed significantly higher protein content, 44 g/100 g, than those from the HP, ~39 g/100 g (*p* < 0.01), a difference attributed to the higher oil yield of the SSP. Similar findings were reported by Roncero et al. (2021) [9] for flours resulting after almond oil extraction with SSP and HP. Rabadán et al. (2018) [1] evaluated flours obtained after pistachio oil extraction using HP across several cultivars, reporting values ranging from 19 to 24 g/100 g fat and 37 to 46 g/100 g protein, within the range of the results in our study for HP processing.

Fibre content was significantly higher (*p* < 0.01) in flours resulting from SSP system compared to HP, with values of 12 g/100 g and 9 g/100 g, respectively. These values were higher than those reported by Rabadán et al. (2017) [40] and Roncero et al. (2021) [9] for partially defatted pistachio and almond flours processed using both SSP and HP systems. This discrepancy may be attributed to the inherent characteristics of the raw material employed in this study: immature pistachio kernels of lower quality and smaller size than those typically intended for consumption. These pistachios had a higher proportion of skin than first-quality pistachios, which likely contributed to the higher fibre content observed. The carbohydrate content was similar in all samples obtained, ~16%, and did not show significant differences (*p* > 0.05) with respect to the extraction system or the pretreatment temperature of the pistachios.

The ash content was also significantly higher (*p* < 0.01) in flours from the SSP system compared to HP, with respective values of ~5.7 g/100 g and ~4.9 g/100 g, also likely associated with the higher oil extraction yields obtained with SSP. The detailed mineral content is presented in Table 3. The flour obtained with SSP system showed higher content of all minerals except iron, with no significant differences (*p* > 0.05), while the pretreatment temperature showed no effect. Among the minerals, potassium (K) was the most abundant, followed by phosphorus (P), magnesium (Mg), and calcium (Ca). Ling et al. (2016) [16] reported similar major minerals and concentrations in SSP partially defatted pistachio flours derived from roasted pistachios. However, in comparison to the study by Burbano and Correa (2021) [41], which examined the mineral composition of defatted walnut flour, the pistachio flours in the present study exhibited higher values for most minerals, with the exception of Mg, Fe, and Zn. Previous research has identified pistachios as a significant source of potassium (K) and phosphorus (P), with a notably higher concentration in the skin of the kernel [42]. In addition to their unsaturated fatty acids, fibre, and proteins, the benefits of pistachios have also been associated with their magnesium content [1]. The high magnesium concentration in the resulting flour is notable as one of the primary nutritional values of this nut, thereby positioning it as a product of significant interest from a nutritional perspective.

#### 3.3.2. Tocopherol Content

Table 3 presents the concentrations of different tocopherol isomers found in the partially defatted pistachio flours obtained in this study. Tocopherols are known for their strong antioxidant activity and their role in disease prevention [43]. These lipid-soluble compounds, comprising α-, β-, γ-, and δ-tocopherols, along with their corresponding tocotrienols, collectively are known as vitamin E. Each isomer possesses distinct physiological functions, making it important to characterise the full vitamin E profile of each food product [15]. In this study, only α-, γ-, and δ-tocopherol were detected in measurable amounts, while β-tocopherol was not detected in the flours. γ-tocopherol was the predominant component for all samples, accounting for approximately 93% of the total tocopherol content, followed by α-tocopherol (~5%) and δ-tocopherol (~2%). Although few studies that have quantified the specific tocopherol composition in flour byproduct obtained from oil extraction of pistachios, several authors have reported γ-tocopherol as the prevalent tocopherol isomer found in nuts [44]. The tocopherol profile of the natural pistachios used for oil extraction followed a similar trend that the flours, with γ-tocopherol being the most abundant (208 mg/kg), followed by α-tocopherol (6.6 mg/kg), and δ-tocopherol (3.75 mg/kg).

Regarding the different samples, the extraction system was the only factor significantly affecting the tocopherol content (*p* < 0.01), with higher levels of all tocopherol isomers found in flours obtained using the HP system compared to those from the SSP system, averaging approximately 156 mg/kg and 111 mg/kg, respectively. The limited literature on tocopherol variation in nut and seed flours after oil extraction suggests that the method of extraction influences antioxidant compound retention. Sarkis et al. (2014) [45] demonstrated that antioxidant content in nut and seed cakes varies depending on the industrial oil extraction process used. Additional studies examining the impact of different heat pretreatments (drying and roasting) on the total tocopherol content of natural cashew nuts [46], oil extracted from pine nuts [43], and nut flours produced through hydraulic pressing [15] showed that higher processing temperatures generally reduced tocopherol levels. In this study, however, tocopherol concentrations were not significantly affected by either the 60 °C pretreatment or by heat exposure during SSP extraction. Since tocopherols contribute to the oxidative stability of oil-rich flours, preserving these compounds enhances both the shelf life and nutritional value of pistachio flours [15]. The higher oil content retained in the partially defatted flours from HP extraction likely accounts for their increased tocopherol levels, supported by a strong positive correlation between fat content and total tocopherol content (r = 0.994, *p* = 0.006).

#### 3.3.3. Colour

Table 3 presents the colour parameters of the partially defatted pistachio flours. All colour parameters were significantly affected by both the extraction system and pretreatment temperature (*p* < 0.001), while their interaction was significant only for *L** and *h* parameters (*p* < 0.001). Flours produced using the HP exhibited significantly higher *L** values, indicating a lighter appearance. For both extraction systems, heating pretreatment at 60 °C resulted in slightly lighter flours than pretreatment at 25 °C. The chromatic coordinates a (red-green) and b (yellow-blue) were positive and more elevated in flours from the SSP extraction, placing their colour in the first quadrant of the chromatic diagram, indicative of red and yellow hues. *a** and *b** exhibited a reduction ranging from 4 to 8% in pistachio flours pretreated at 60 °C compared to those pretreated at 25 °C. The *h* values ranged from ~68 in SSP flours (more reddish) to ~76 in HP flours (more yellowish). The *C** values, representing colour saturation, were on average 32% higher in SSP flours, suggesting more vivid and intense colours. These findings align with those of Labuckas et al. (2014) [47], who reported darker and redder colours for walnut flours defatted with SSP compared to those of HP. This is attributed to localised temperature increases during SSP processing, reaching up to 80 °C, due to the mechanical pressure and friction of the screw. Such conditions promote Maillard and caramelisation reactions from the components present in the flours, altering the colour [47]. In addition, the heating of pistachios from 60 °C initiates the thermal degradation of their natural pigments, primarily chlorophylls and carotenoids, leading to alterations in their chromatic profile [43]. In contrast, the higher residual oil and lower protein and carbohydrate content, and lower temperatures achieved during processing in HP flours may limit these reactions, contributing to their lighter, yellowish hues.

### 3.4. Techno-Functional Properties of Flours

#### 3.4.1. Hydration Properties

The findings regarding the hydration properties of defatted pistachio flours are presented in Figure 2a,b.

The water absorption capacity (WAC) of the flours was significantly affected by the oil extraction system (*p* < 0.001), but not by the pretreatment temperature (*p* > 0.05). However, a significant interaction between the extraction system and temperature was observed (*p* < 0.05). WAC values were higher in samples obtained using the HF system, with PF-HP25 showing the highest value (2.75 g/g). In contrast, samples obtained using the SSP system showed lower WAC values (~2 g/g), without significant differences between the two pretreatment temperatures. The primary contributors to WAC in flours are proteins and carbohydrates, as both molecules possess polar or charged side chains that are hydrophilic, thereby facilitating water absorption [48,49]. The HP system, distinguished as a “cold” extraction method, likely preserves greater quantities of hydrophilic protein constituents in the resultant flours, especially when samples are processed at lower or controlled temperatures. This preservation may explain the enhanced WAC observed in HP-derived flours.

The water absorption index (WAI) and swelling power (SP) (Figure 2a) were higher in flour samples obtained with a pretreatment at 25 °C, regardless of the extraction system, with average values of ~5.7 g/g for WAI and ~7.0 g/g for SP. In contrast, samples treated at 60 °C exhibited lower values and showed differences among systems, with 4.73 g/g (HP) and 4.27 g/g (SSP), and SP of 6.09 g/g (HP) and 5.64 g/g (SSP). The observed variations in the samples, attributable to the extraction system and pretreatment temperature, similar to WAC, may be associated with the improved preservation of the protein structure with HP and milder pretreatment conditions. In contrast, increasing the pretreatment temperature to 60 °C and the shearing effect during oil extraction with the SSP could impact protein conformation and affects its polarity and hydrophobicity. These structural changes may result in the weakening of hydrogen bonds, thereby affecting protein interactions with water during sample heating [50].

The water solubility index (WSI) (Figure 2b) increased with higher pretreatment temperature, rising by 25% in SSP and 22% in HP when comparing flours from pistachios pretreated at 60 °C versus 25 °C. For the same pretreatment temperature, WSI values were consistently higher in SSP-derived flours, by 6% at 25 °C and 9% at 60 °C, compared to those obtained via HP. WSI reflects the presence of soluble compounds within the studied matrix, whereas WAC is primarily influenced by the insoluble fraction and its hydration capacity [47]. Labuckas et al. (2014) [47] reported analogous findings in flours derived from nut defatting using both screw and hydraulic presses. Those authors attributed the observed differences to the greater degree of protein alteration in samples obtained by SSP. This alteration was attributed to the mechanical friction and elevated temperatures experienced during the process of pressing, which likely result in an increased soluble fraction in the flours. In addition, the pretreatment would have facilitated the potential structural modifications of the proteins, thereby enhancing the soluble fraction of the flours.

#### 3.4.2. Emulsifying Properties

The emulsifying properties of emulsion activity (EA) and emulsion stability (ES) of the flours under investigation are presented in Figure 2c. The oil extraction system significantly affected the emulsifying properties (*p* < 0.05), while the pretreatment temperature and its interaction with the extraction method had no significant effect (*p* > 0.05). EA values slightly increase with the use of higher pretreatment temperature and with the SSP system, although a statistically significant difference was only observed between FP-SSP-60 and FP-HP25. In terms of emulsion stability after heating at 80 °C, the flours obtained using SSP displayed notable stability (35.2% for FP-SSP-60 and 33.3% for FP-SSP-25), whereas those produced via HP showed negligible stability. The EA of the samples presented similar values, although slightly lower, than those obtained in the studies of Fernández-López et al. (2018) [21] and Ling et al. (2016) [16], for defatted chia flour and fully defatted pistachio flour, respectively. Sanchiz et al. (2019) [14] reported that the application of various heat treatments, such as boiling and autoclaving, to defatted pistachio, cashew, and chestnut flours resulted in a reduction in EA and ES. However, in the present study, the pretreatment temperature of natural pistachios did not exert any significant effect on these techno-functional characteristics (*p* > 0.05). EA and ES are primarily influenced by the protein and fat content, as both are powerful emulsifying agents [21]. In addition, the emulsifying properties of vegetable flours are influenced not only by their protein content but also by the characteristics of the proteins themselves, such as the proportion of hydrophilic amino acids, which contribute to the reduction in surface tension at a liquid–liquid interface, and hydrophobic amino acids, which serve as binding sites [16,21]. Possible differences in the protein characteristics of the flour samples obtained from the two extraction systems used in this study, such us denaturation or unfolding, could justify the different ES values obtained. Previous studies on partially defatted hemp flours obtained by screw pressing confirmed the thermal stability of the protein in the samples, showing that the flours do not undergo protein denaturation [20]. However, an increase in temperature during pre-treatment and subsequent extraction with this press could cause the protein structure to unfold, which would have a positive effect on the flour’s emulsion stability.

#### 3.4.3. Foaming Properties

Figure 2d shows the foaming properties of foaming capacity (FC) and foaming stability (FS) of the studied pistachio flours. A higher pretreatment temperature slightly increased FC, being only significant for the HP system (64 mL for PF-HP60 and 61.5 for PF-HP25). The FS ranged from 44 to 47%, with no significant differences among samples (*p* > 0.05). The FC values observed for the fully defatted pistachio flour samples were in agreement with those reported by Jitngarmkusol et al. (2008) [49] and Ling et al. (2016) [16] for fully defatted macadamia nut and pistachio nut flours, and were significantly higher than those found by Sanchiz et al. (2019) [14] for defatted chestnut and cashew flours. Previous studies have demonstrated that heat treatments applied to different seeds and nuts have been the cause of the reduction, and even elimination, of the foaming capacity of flours. Roasting can lead to modifications or losses in the structural and conformational properties of proteins, and may enhance protein–protein interactions, resulting in the formation of aggregates that adversely affect foaming [14,16]. The natural character of the samples used in this study, in conjunction with the mild conditions of the pretreatments and extraction processes to which they were subjected, may account for the absence of significant differences in the results obtained for the foaming properties of the partially defatted pistachio flours.

## 4. Conclusions

This study highlights the potential of partially defatted pistachio flours, derived as by-products of oil extraction, as valuable ingredients for the food industry. Their techno-functional and nutritional properties varied notably depending on the oil extraction method (hydraulic press or single-screw press) and the pretreatment temperature applied to the raw material (25 °C or 60 °C). These variations were largely attributed to differences in composition, particularly residual fat content, and to the preservation or alteration of protein structures during processing.

The HP system likely preserved a greater proportion of hydrophilic proteins, enhancing the WAC of the resulting flours. These flours also exhibited a lighter colour and higher tocopherol content, attributes beneficial for use in food matrices with specific organoleptic requirements or where enhanced oxidative stability is desired. In contrast, flours obtained using the SSP system were characterised by a higher WSI, EA, and ES, key properties for use in formulations such as creams, sauces, and meat products. The notable emulsion stability suggests further application potential in whipped batters and baked goods, where emulsifying and stabilising agents are needed. In addition, SSP-derived flours exhibited a remarkable nutritional composition, due to their higher oil extraction yield, leading to high contents of protein, fibre, ash, and minerals. While pretreatment temperature had a more limited effect overall, milder conditions (25 °C) contributed to higher WAI and SP, enhancing their potential applications in the formulation of bakery and pastry products, especially in gluten-free formulations. The potential use of these flours in various food applications would require a specific study of the operating conditions during the extraction process, according to the technological requirements of each formulation.

The pistachio oils extracted exhibited a lipid profile characterised by a high content of monounsaturated and polyunsaturated fatty acids, particularly oleic acid (*ω-9*), making them attractive for both culinary and non-food applications, such as cosmetics and pharmaceuticals.

In summary, this work underscores the feasibility of converting low-commercial-quality pistachios into high-value ingredients: pistachio oils and partially defatted flours. These ingredients, especially the flour by-products, possess a promising combination of nutritional and techno-functional properties suitable for incorporation into a wide range of food products. These findings support the valorisation of agro-industrial by-products and align with the principles of circular economy, offering a pathway to reduce food waste while enriching food formulations with natural and functional ingredients. Future research should focus on applying these flours in food product development, particularly in the formulation of bakery and pastry items, in which their functional and nutritional properties could be fully exploited.

## Figures and Tables

**Figure 1 foods-14-02722-f001:**
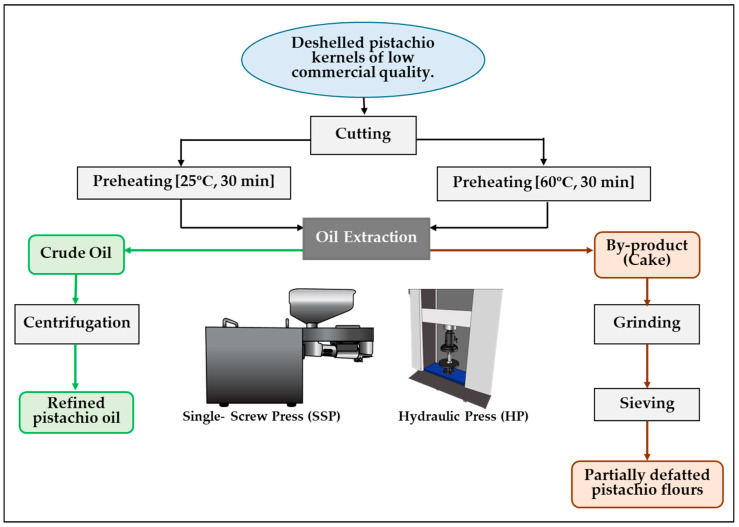
Schematic representation of the processing carried out on natural deshelled kernels obtained from pistachios of low commercial quality.

**Figure 2 foods-14-02722-f002:**
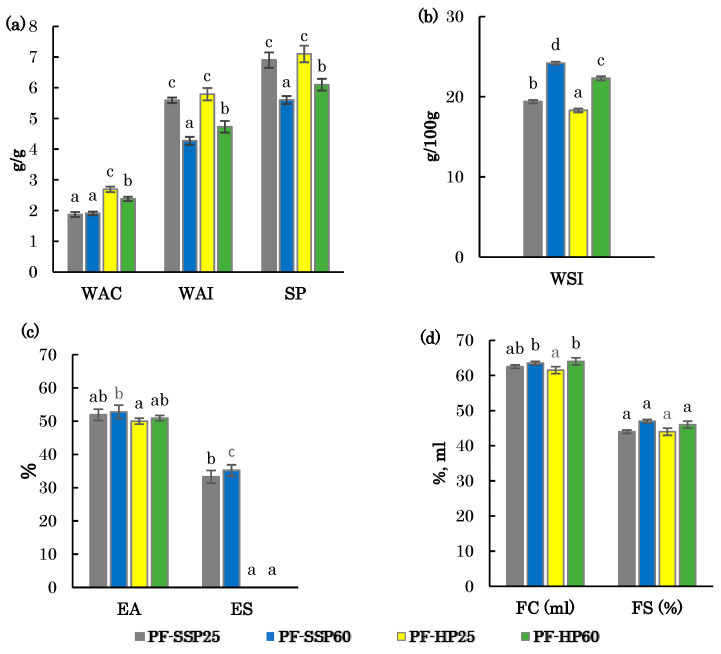
Techno-functional properties of defatted flours obtained using a single-screw press (SSP) and hydraulic press (HP) from natural pistachios pretreated at 25 °C and 60 °C. PF-SSP25, PF-SSP60, PF-HP25, and PF-HP60: Flours of pistachios treated at 25 and 60 °C using SSP and HP systems, respectively. (**a**) WAC: water absorption capacity; WAI: water absorption index; SP: swelling power; (**b**) WSI: water solubility index; (**c**) EA: emulsifying activity; ES: emulsion stability (**d**) FC: foaming capacity; FS: foaming stability; All parameters are referred to dry matter. The error bars represent the standard deviation. Bars with different letters for a same parameter represent significant differences among samples (*p* < 0.05).

**Table 1 foods-14-02722-t001:** Extraction yields of pistachios and oil characterisation.

Parameters	Samples					*p*-Values		
	PO-SSP25	PO-SSP60	PO-HP25	PO-HP60	SE	F1	F2	F1 × F2
Extraction yield (g/100 g)								
Oil	40.1 b	39.7 b	23.4 a	25.1 a	0.9	***	ns	ns
Cake	59 a	60 a	76 b	75 b	1	***	ns	ns
Colour								
*L**	38 b	33 a	45 c	45 c	1	**	ns	*
*a**	−6.4 b	−5.4 c	−9.3 a	−4.1 d	0.1	ns	**	***
*b**	91 b	81 a	99 c	99 c	2	**	ns	*
*C**	91 b	82 a	99 c	100 c	2	**	ns	*
h	94.0 b	93.8 b	95.4 c	92.4 a	0.3	ns	*	**
Free acidity (% oleic acid)	0.6 a	0.6 a	0.6 a	0.7 a	0.1	ns	ns	ns

PO-SSP25 and PO-SSP60: Oils extracted from pistachios pretreated at 25 °C and 60 °C, respectively, using the single-screw press (SSP). PO-HP25 and PO-HP60: Oils extracted from pistachios pretreated at 25 °C and 60 °C, respectively, using the hydraulic press (HP). *L**: lightness; *a** and *b**: colour coordinates; *C**: chroma; h: hue; SE: pooled standard error obtained ANOVA. Values in the same row with different letters differ significantly at *p* < 0.05. Analysis of variance and significance of extraction method (SSP or HP) (F1) and pretreatment temperature (25 °C or 60 °C) (F2), and their interaction (F1 × F2): *** *p* < 0.001; ** *p* < 0.01; * *p* < 0.05; ns: non-significant.

**Table 2 foods-14-02722-t002:** Fatty acid profile of pistachio oils obtained by single-screw press and hydraulic press with pretreatment at 25 °C.

	PO-SSP25	PO-HP25	SE	*p*-Value
Palmitic, C16:0	14 a	14 a	1	ns
Stearic, C18:0	0.9 a	0.9 a	0.2	ns
Arachidic, C20:0	0.11 a	0.11 a	0.02	ns
Behenic, C22:0	0.11 a	0.11 a	0.02	ns
Lignoceric, C24:0	0.05 a	0.06 a	0.01	ns
Palmitoleic, C16:1	1.5 a	1.5 a	0.3	ns
Vaccenic, C18:1n7	2.6 a	2.8 a	0.5	ns
Oleic, C18:1n9	49 a	47 a	4	ns
Gondoic, C20:1n9	0.3 a	0.3 a	0.1	ns
Linoleic, C18:2n6	30 a	31 a	2	ns
α-Linolenic, C18:3n3	0.8 a	0.9 a	0.2	ns
SFA	15.1 a	15.6 a	0.9	ns
MUFA	54 a	51 a	5	ns
PUFA	31 a	32 a	4	ns
TFA	0.20 a	0.07 b	0.02	*
*ω-3*	0.8 a	0.9 a	0.1	ns
*ω-6*	30 a	31 a	4	ns
*ω-9*	49 a	48 a	4	ns

PO-SSP25 and PO-HP25: Oils extracted from pistachios pretreated at 25 °C using the single-screw press (SSP) and the hydraulic press (HP), respectively. Fatty acids expressed as g/100 g oil. SFAs: saturated fatty acids; MUFAs: monounsaturated fatty acids; PUFAs: polyunsaturated fatty acids; TFAs: trans fatty acids; *ω-3*: omega-3 fatty acids; *ω-6*: omega-6 fatty acids; *ω-9*: omega-9 fatty acids. Values in the same row with different letters differ significantly at *p* < 0.05. Analysis of variance and significance of extraction method (SSP or HP): *** *p* < 0.001; ** *p* < 0.01; * *p* < 0.05; ns: non-significant.

**Table 3 foods-14-02722-t003:** Nutritional composition and colour of flour by-products obtained after oil extraction from pistachios.

Parameters	Samples					*p*-Values		
	PF-SSP25	PF-SSP60	PF-HP25	PF-HP60	SE	F1	F2	F1 × F2
Proximal composition (g/100 g)								
Moisture	3.9 b	3.4 a	4.8 d	4.6 c	0.1	***	***	***
Carbohydrates	17 a	16 a	16 a	15 a	2	ns	ns	ns
Fat	17.4 a	18.3 a	28.5 b	28.1 b	0.8	***	ns	*
Protein	44 b	44 b	38 a	39 a	3	**	ns	ns
Fibre	12 b	12 b	8 a	8 a	2	**	ns	ns
Ash	5.6 b	5.8 b	4.9 a	4.8 a	0.2	**	ns	ns
Mineral composition (mg/100 g)								
Potassium (K)	1714 b	1691 b	1495 a	1499 a	21	***	ns	ns
Phosphorus (P)	1020 b	993 b	885 a	941 ab	25	*	ns	ns
Magnesium (Mg)	213 c	201 bc	181 a	194 b	5	**	ns	*
Calcium (Ca)	186 c	179 c	160 a	174 b	4	**	ns	*
Iron (Fe)	5.3 a	5.1 a	4.8 a	4.6 a	0.7	ns	ns	ns
Zinc (Zn)	3.1 c	2.9 b	2.6 a	2.9 b	0.1	**	ns	*
Tocopherols (mg/kg):								
γ-tocopherol	103 a	103 a	143 b	146 b	10	**	ns	ns
α-tocopherol	5.9 a	6.0 a	8.2 b	8.7 b	0.6	**	ns	ns
δ-tocopherol	2.3 b	2.2 b	3.1 a	3.4 a	0.2	**	ns	ns
Total tocopherols	111 a	111 a	154 b	158 b	10	**	ns	ns
Colour								
*L**	56.4 a	57.8 b	75.0 c	78.2 d	0.1	***	***	***
*a**	16.1 d	15.2 c	8.3 b	7.2 a	0.1	***	***	ns
*b**	39.1 d	37.6 c	31.1 b	29.6 a	0.1	***	***	ns
*C**	42.3 d	40.5 c	32.1 b	30.4 a	0.1	***	***	ns
h	67.6 a	68.0 b	75.0 c	76.3 d	0.1	***	***	***

PF-SSP25 and PF-SSP60: Flour by-products obtained after oil extraction from pistachios pretreated at 25 °C and 60 °C, respectively, using the single-screw press (SSP). PF-HP25 and PF-HP60: Flour by-products obtained after oil extraction from pistachios pretreated at 25 °C and 60 °C, respectively, using the hydraulic press (HP). *L**: lightness; *a** and *b**: colour coordinates; *C**: chroma; h: hue; SE: pooled standard error obtained ANOVA. Values in the same row with different letters differ significantly at *p* < 0.05. Analysis of variance and significance of extraction method (SSP or HP) (F1) and pretreatment temperature (25 °C or 60 °C) (F2), and their interaction (F1 × F2): *** *p* < 0.001; ** *p* < 0.01; * *p* < 0.05; ns: non-significant.

## Data Availability

The data presented in this study are available on request from the corresponding author.

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
