# Peer review of "Oil Extraction Systems Influence the Techno-Functional and Nutritional Properties of Pistachio Processing By-Products"

_foods, 2025, doi:10.3390/foods14152722_

Round 1
Reviewer 1 Report
Comments and Suggestions for Authors
The publication entitled "Oil Extraction Systems Influence the Techno-Functional and Nutritional Properties of Pistachio Processing By-Products" addresses the highly relevant topic of utilizing by-products from the food industry. The authors analyze the impact of different oil extraction methods on the technological and nutritional properties of pistachio processing residues. The study fits into the growing interest in sustainable development and the search for new applications for secondary raw materials in the food industry. The research presented in the article may be of significant importance both for scientists and practitioners involved in food processing. The text makes a valuable contribution to the development of knowledge on the rational use of by-products.
The submitted publication presents methods for obtaining oil from pistachio nuts as well as techniques for producing flour from the residual material. The raw material used in these studies comprises pistachio nuts that do not meet quality standards (e.g., those that are too small, broken, etc.). In the oil extraction process, the authors employed two types of presses: hydraulic and screw presses.
Although the pressing process itself does not constitute an innovative solution, both the resulting oil and flour are noteworthy due to their unique physicochemical and organoleptic properties. The analyses conducted by the authors to thoroughly characterize the composition were performed correctly. Standardized measurement methods, widely used in analogous research, were applied—for example, the protein content in the raw material was determined using the Kjeldahl method, and the fat content was assessed by solvent extraction using the Soxhlet method (AACC Method 30-25.01).
I believe that the article, in its current form, can be accepted for publication in the Foods. However, I can suggest some minor changes:
- There is a complete lack of information regarding measurement errors. For instance, in the characterization of the raw material, the number of repetitions for each determination and the standard deviation of the measurements were not reported; only approximate values were provided. Furthermore, the authors did not specify how the representative sample of nuts was selected. There is also no information, even approximate, regarding the storage time of the nuts prior to the oil extraction process.
In summary, while the experimental methodology is sound and the use of standardized analytical techniques is commendable, the manuscript would benefit from greater transparency in reporting measurement uncertainties and sample selection procedures
- The article provides particular value through its technological description of the process of obtaining oil from pistachio nuts. However, the oil pressing procedure itself, as well as the analytical methods employed, do not constitute a novelty in and of themselves. The oil analysis was conducted correctly, yet in a rather limited scope—restricted to the fatty acid profile and color analysis using the CIE LAB scale. It would be advisable to include, at a minimum, the sterol and tocopherol profiles, as well as to consider factors such as the impact of pressing temperature on oil quality. The analysis of the resulting flour was carried out comprehensively, and I have no remarks in this regard.
- In my opinion, the authors have appropriately selected publications to accurately define the research problem. In most cases, these are articles published after 2020 and are closely related to the main topic of the manuscript. However, it should be noted that some of the cited works date back to the 1990s.
I kindly request an update of the literature, as many of the sources provided are from the 1990s and the early 2000s. Incorporating more recent references would strengthen the scientific foundation of the manuscript and ensure it reflects the current state of research in the field. - Please provide a diagram or photo of the presses used for oil extraction. This could be of interest to the readers of the publication.
- In the case of Table 1, it is worth considering including information about the percentage changes of the individual LAB parameters. This would be a more accessible format for the reader.
The conclusions presented in the article are well-aligned with the reported results and constitute an interesting summary. I believe the Authors could outline a plan for future research, as the topic is highly engaging and holds significant potential for further exploration.
Author Response
Responses to Comments / Requests raised by the Reviewers
We would like to thank the reviewers for their valuable comments, which definitely have helped us to improve the quality of the manuscript. In revising the text of the paper, we have taken into account all the reviewers’ comments and suggestions. The changes made in the manuscript (all highlighted in the text), as well as our responses to the comments/queries raised by the reviewers, are presented below.
Reviewer 1
The publication entitled "Oil Extraction Systems Influence the Techno-Functional and Nutritional Properties of Pistachio Processing By-Products" addresses the highly relevant topic of utilizing by-products from the food industry. The authors analyze the impact of different oil extraction methods on the technological and nutritional properties of pistachio processing residues. The study fits into the growing interest in sustainable development and the search for new applications for secondary raw materials in the food industry. The research presented in the article may be of significant importance both for scientists and practitioners involved in food processing. The text makes a valuable contribution to the development of knowledge on the rational use of by-products.
The submitted publication presents methods for obtaining oil from pistachio nuts as well as techniques for producing flour from the residual material. The raw material used in these studies comprises pistachio nuts that do not meet quality standards (e.g., those that are too small, broken, etc.). In the oil extraction process, the authors employed two types of presses: hydraulic and screw presses.
Although the pressing process itself does not constitute an innovative solution, both the resulting oil and flour are noteworthy due to their unique physicochemical and organoleptic properties. The analyses conducted by the authors to thoroughly characterize the composition were performed correctly. Standardized measurement methods, widely used in analogous research, were applied—for example, the protein content in the raw material was determined using the Kjeldahl method, and the fat content was assessed by solvent extraction using the Soxhlet method (AACC Method 30-25.01).
I believe that the article, in its current form, can be accepted for publication in the Foods.
RESPONSE: Thank you for your kind comments and good consideration of our work.
However, I can suggest some minor changes:
1. There is a complete lack of information regarding measurement errors. For instance, in the characterization of the raw material, the number of repetitions for each determination and the standard deviation of the measurements were not reported; only approximate values were provided. Furthermore, the authors did not specify how the representative sample of nuts was selected. There is also no information, even approximate, regarding the storage time of the nuts prior to the oil extraction process.
In summary, while the experimental methodology is sound and the use of standardized analytical techniques is commendable, the manuscript would benefit from greater transparency in reporting measurement uncertainties and sample selection procedures
RESPONSE: Thank you for reviewer’s comments. Regarding measurement uncertainty, we have now included the number of replicates and standard deviations for the characterisation of the raw material (see Lines 105-106 and 114). For the other analyses, pooled standard error (SE) is presented in the tables instead of individual standard deviations. To justify this approach, we first confirmed the homogeneity of variance among the samples using ANOVA. This confirmed that the use of pooled SE was statistically valid and did not compromise the representation of variability in the data. This method also facilitates clearer visualisation and comparison of values across samples and treatments.
Regarding the selection and processing of the pistachio sample, we have clarified that the nuts used corresponded to a representative batch provided by the company Pistacyl. The pistachios were dried post-harvest to stabilise their moisture content below 5%, shelled, and subsequently stored under modified atmosphere (100% N₂) until analysis. Additional details regarding the preprocessing and storage time and conditions have been added (see Lines 100-104).
2. The article provides particular value through its technological description of the process of obtaining oil from pistachio nuts. However, the oil pressing procedure itself, as well as the analytical methods employed, do not constitute a novelty in and of themselves. The oil analysis was conducted correctly, yet in a rather limited scope—restricted to the fatty acid profile and color analysis using the CIE LAB scale. It would be advisable to include, at a minimum, the sterol and tocopherol profiles, as well as to consider factors such as the impact of pressing temperature on oil quality. The analysis of the resulting flour was carried out comprehensively, and I have no remarks in this regard.
RESPONSE: Thank you for reviewer’s suggestions. We would like to clarify that the primary objective of this study is to characterise the partially defatted pistachio flours, focusing mainly on their techno-functional and nutritional properties. In this context, the effect of processing temperature on the nutritional properties of the flours was indirectly assessed by measuring the tocopherol content of the residual oil in the flours (see Table 3). While the oil analysis itself was limited to fatty acid profile and colour, the tocopherol content in the flour reflects important aspects of oil quality affected by processing. We acknowledge that including sterol profiles and a detailed assessment of pressing temperature effects on oil quality would be valuable and will be considered in future studies more specifically focused on the oil fraction.
3. In my opinion, the authors have appropriately selected publications to accurately define the research problem. In most cases, these are articles published after 2020 and are closely related to the main topic of the manuscript. However, it should be noted that some of the cited works date back to the 1990s.
I kindly request an update of the literature, as many of the sources provided are from the 1990s and the early 2000s. Incorporating more recent references would strengthen the scientific foundation of the manuscript and ensure it reflects the current state of research in the field.
RESPONSE: Thank you for your valuable comment regarding the literature cited. After a thorough review, we confirm that the manuscript does not include references from the 1990s. The majority of cited works date from the mid-2000s onward, reflecting a period of growing scientific interest in pistachio and nut-related research. Notably, many references are from the last five years (2020–2025), providing an up-to-date information on pistachio oil characterization, by-product utilization, nutritional properties, and extraction technologies. Therefore, we believe that the selected literature offers a solid and current scientific foundation for our research. Nevertheless, a new reference from 2020 has been added to support the discussion of the results obtained (see Lines 245 and 702-704).
4. Please provide a diagram or photo of the presses used for oil extraction. This could be of interest to the readers of the publication.
RESPONSE: Thank you for your suggestion. We have included a detailed diagram of the oil extraction process used in this study as Figure 1 to help readers better understand the methodology applied.
5. In the case of Table 1, it is worth considering including information about the percentage changes of the individual LAB parameters. This would be a more accessible format for the reader.
RESPONSE: Thank you for reviewer’s suggestion. In this study, a multifactorial design was used to evaluate the effect of two factors, the extraction system (hydraulic press or single-screw press) and the pretreatment temperature (25 °C or 60 °C), on the colour parameters of the flours. Therefore, there is no single reference treatment against which percentage changes can be meaningfully calculated. Instead, statistical analysis was carried out to determine the significance of each factor and their interaction on the colour coordinates. However, we appreciate the reviewer’s perspective and will consider including percentage differences in future work where appropriate reference points are available.
6. The conclusions presented in the article are well-aligned with the reported results and constitute an interesting summary. I believe the Authors could outline a plan for future research, as the topic is highly engaging and holds significant potential for further exploration.
RESPONSE: Thank you for your positive evaluation and valuable suggestion. We agree that this line of research offers interesting possibilities for further exploration, and we have included some remarks in the conclusions section as suggested (see Lines 591-594).

Reviewer 2 Report
Comments and Suggestions for Authors
In this manuscript, oil was extracted from the low quality Pistachios with different pressing methods and the obtained cakes were also used for their functional properties. It is very interesting subject and well performed research, particularly the functional properties of the obtained flour were novel and useful for the future applications.
Line 21, as oil extracted by different methods and also after thermal treatments, please include which treatment had the high extraction yield.
It would be interesting to have a Figure in method section to show and summarize whole works as flowchart.
Please see the Table 1 for oil content of obtained flour, as the oil contents are different, it affected all other factors, most importantly effects on the mineral. As minerals are very stable during different food processing, so they are not affected by thermal processing and so on. It means that differences in the mineral content or other components are as results of dilution by oil remained in cake. Please include this matter and issue in discussion section as well.
The main reason that thermal pre-treatment could not affect the oil content or other parameters was due to the lower temperature and lower duration of treatment (30 min), which could be included in the discussion part for more interpretation. Please see below paper for more detail:
Effect of roasting and microwave pre-treatments of Nigella sativa L. seeds on lipase activity and the quality of the oil.Food Chemistrty, 2019, 274, pp. 480–486
Author Response
Responses to Comments / Requests raised by the Reviewers
We would like to thank the reviewers for their valuable comments, which definitely have helped us to improve the quality of the manuscript. In revising the text of the paper, we have taken into account all the reviewers’ comments and suggestions. The changes made in the manuscript (all highlighted in the text), as well as our responses to the comments/queries raised by the reviewers, are presented below.
Reviewer 2
In this manuscript, oil was extracted from the low-quality Pistachios with different pressing methods and the obtained cakes were also used for their functional properties. It is very interesting subject and well performed research, particularly the functional properties of the obtained flour were novel and useful for the future applications.
RESPONSE: Thank you for your kind comments and positive evaluation of our work.
1. Line 21, as oil extracted by different methods and also after thermal treatments, please include which treatment had the high extraction yield.
RESPONSE: Thank you for your suggestion. We have added a sentence in the abstract (see Lines 21–22) indicating which treatment achieved the highest oil extraction yield, as discussed in Section 3.1 of the results and discussion.
2. It would be interesting to have a Figure in method section to show and summarize whole works as flowchart.
RESPONSE: Thank you for your suggestion. To improve the clarity of the work, a flowchart summarising the experimental procedure has been included in the methods section (see Figure 1).
3. Please see the Table 1 for oil content of obtained flour, as the oil contents are different, it affected all other factors, most importantly effects on the mineral. As minerals are very stable during different food processing, so they are not affected by thermal processing and so on. It means that differences in the mineral content or other components are as results of dilution by oil remained in cake. Please include this matter and issue in discussion section as well.
RESPONSE: Thank you for your observation. To complement the already included discussion, a general discussion about the influence of residual oil content on the composition of the flour has been added (see Lines 355-357, 360-361 and 389-390)
4. The main reason that thermal pre-treatment could not affect the oil content or other parameters was due to the lower temperature and lower duration of treatment (30 min), which could be included in the discussion part for more interpretation. Please see below paper for more detail:
Effect of roasting and microwave pre-treatments of Nigella sativa L. seeds on lipase activity and the quality of the oil (Food Chemistrty, 2019, 274, pp. 480–486)
RESPONSE: Thank you for your suggestion. We have addressed this point in the discussion section (see Lines 242-245), including the reference suggested by the reviewer to support the statement.
